# Effect of Saline Water for Drip Irrigation on Microbial Diversity and on Fertility of Aeolian Sandy Soils

**Xiangxiang Yu [1,2,3], Zhengzhong Jin [1,2,3,*] and Haifeng Wang [1,2,3]**

1   National Engineering Technology Research Center for Desert-Oasis Ecological Construction, Xinjiang Institute of Ecology and Geography, Chinese Academy of Sciences, Urumqi 830011, China; yuxiangxiang@ms.xjb.ac.cn (X.Y.); wanghf@ms.xjb.ac.cn (H.W.)
2   Mosuowan Desert Research Station, Xinjiang Institute of Ecology and Geography, Chinese Academy of Sciences, Shihezi 832000, China
3   The Taklimakan Desert Research Station, Xinjiang Institute of Ecology and Geography, Chinese Academy of Sciences, Korla 841000, China
*   Correspondence: jinzz@ms.xjb.ac.cn

**Abstract:** Saline water is widely distributed in the arid environment and sometimes represents the only source of irrigation water to restore and reconstruct vegetation. However, the effects of saline water on the bacterial diversity and fertility level of aeolian sandy soil are not well understood. In this study, we investigated a vegetation belt along the Tarim Desert Highway that has been constructed as a windbreak and consists of desert shrubs and was irrigated with saline water at six levels of salinity along the Tarim Desert Highway. The bacterial diversity was studied using Biolog Eco, a phospholipid fatty acid analysis, and a polymerase chain reaction–denaturing gradient gel electrophoresis, and the soil fertility was calculated and expressed as the integrated fertility index. The soil bacterial diversity (in terms of carbon metabolism, genes, and fatty acid species) was significantly affected by the level of salinity, and the microbial activity was low under high salinity. Fertility was also markedly affected by the degree of salinity and by the depth of soil, being lower at higher salinity levels and in the top layer (0–5 cm), and was also correlated to both the metabolic diversity index of soil microorganisms and the diversity index of fatty acids of soil microorganisms. The genetic diversity index of soil microorganisms shared a polynomial relation with fertility and contributed to it positively and significantly. Therefore, using less saline water for drip irrigation could avoid salt accumulation in soil and arrest its compaction, promote the formation of soil aggregates and the build-up of nutrients, and increase microbial activity, thus playing a crucial role in promoting the circulation, conversion, and utilization of nutrients in aeolian sandy soils and improving the soil quality. The judicious use of saline water, therefore, deserves serious consideration in irrigation practices.

**Keywords:** extremely arid region; saline water irrigation; biological activity; soil fertility

## 1. Introduction

Soil fertility is generally evaluated in terms of its physical, chemical, and biological properties [1], and it is very important to evaluate the fertility of soils [2]. A commonly used method for expressing soil quality in quantitative terms is the integrated fertility index (IFI) of soil [3]. Changes in the numbers of soil microorganisms and their metabolic functions can be used as sensitive indicators of the short-term and long-term changes in soil fertility [4]. Fertile soils have stable microbial populations (stable in terms of numbers and in the composition of microbial communities) and a high level of microbial activity [5]. Soil microbiome is a potential and sensitive indicator of soil fertility [6]. It is important to study the effects of changes in soil fertility on the microbial diversity.

The Tarim Desert Highway shelterbelt across the Taklimakan Desert in China is a piece of ecological engineering to ensure the safe operation of the Tarim Desert Highway.



However, the harsh natural conditions, such as extreme temperatures, high salinity, and sandstorms, are serious threats to the shelterbelt. The Taklimakan Desert has an extremely arid climate and a serious lack of surface water. The abundant groundwater with different salt content (approximately 2–29 $g/L^{-1}$) is the main source of irrigation water for the construction of the shelterbelt in the hinterland of the desert. Although salt water irrigation can cause soil salt accumulation, the species of tree planted in the shelterbelt are three shrubs named *Haloxylon ammodendron* (C. A. Mey.) Bunge, *Calligonum mongolicum* Turcz., and *Tamarix chinensis* Lour., which are resistant to salt and alkali. Therefore, the Tarim Desert Highway shelterbelt located in the hinterland of the Taklimakan Desert can only be kept alive by pumping salt water underground for common drip irrigation. The present study aims to examine the pattern of changes in the fertility of the aeolian sandy soils that surround the highway in response to irrigation with saline water to obtain useful insights for sustainable development of the shelterbelt.

Soil microorganisms and soil quality in arid areas have been studied extensively. The results suggest that microorganisms are an important source of available nitrogen in woodland soils of the desert ecosystem [7] and that they improve soil fertility once the soil is brought under cultivation [8]. The relationship between soil microorganisms and soil quality was investigated by Shao et al. [9] in the shrub forest consisting mostly of *Artemisia ordosica* Krasch in the Kubuqi Desert; by Wang et al. [10] in the improved semi-mobile sands of Ephedra along artemisia ordosicaartemisia ordosicaartemisia ordosicaartemisia ordosicathe southern edge of the Mu Us Sandland; and by Cao et al. [11] in the sand dunes that support *Caragana microphylla* Lam. in the Horqin Sandland in Inner Mongolia. The researchers found that the roots of the sand-fixing plants provide soil microorganisms with nutrient-rich metabolites and promote the development of the rhizosphere; at the same time, the metabolic activities of the rhizospheric microorganisms influence the movement of nutrients in the soil and their uptake by plants. This symbiotic relationship thus contributes to the formation of aeolian sandy soils [12].

However, most of the research mentioned above on soil microorganisms focussed on non-irrigated woodlands and woodlands irrigated with fresh water in arid and semi-arid areas, whereas the evaluation of aeolian sandy soils of the shelterbelt irrigated with saline water received little attention.

The present study used conventional methods to determine the bacterial diversities of these sandy soils irrigated with saline water. The following hypotheses were addressed: (i) saline water is significantly altering the soil bacterial diversity and soil fertility in the shelterbelt and (ii) the soil bacterial diversity and soil fertility are depending on irrigation water salt concentrations and on soil depth. The findings will provide a theoretical underpinning for the management of the shelterbelt along the Tarim Desert Highway.

## 2. Materials and Methods

### 2.1. Experimental Design and Sampling

The areas that border the Taklimakan Desert are characterized by an extremely arid climate, scarcity of surface water resources, highly saline groundwater, strong winds, encroachment of sand, and infertile soil. The Tarim Desert Highway shelterbelt across the Taklimakan Desert was established in 2003. Three desert shrub species with a good tolerance to drought and salinity, namely *Haloxylon ammodendron* (C. A. Mey.) Bunge, *Calligonum mongolicum* Turcz., and *Tamarix chinensis* Lour., were planted as mixed rows with a spacing of 1 m × 1 m to form a shelterbelt that was 72–78 m wide and 436 km long.

Three desert shrub species with a good tolerance to drought and salinity, namely *Haloxylon ammodendron* (C. A. Mey.) Bunge, *Calligonum mongolicum* Turcz., and *Tamarix chinensis* Lour., received drip irrigation to the shelterbelt with saline water. There are great spatial differences in the groundwater salinity along the Tarim Desert Highway shelterbelt as the plantation times on some sections of the shelterbelt are different (Table 1). To avoid the influence of stand age and tree species on the soil property, we selected sampling sites with the same stand age (planted in 2003) and tree species (only *Haloxylon ammodendron* (C.

A. Mey.) Bunge). At each site, six plants of *Haloxylon ammodendron* (C. A. Mey.) Bunge were selected at random and the soil was sampled from points close to the roots [13] to represent four depths, namely 0–5 cm ($D_1$), 5–15 cm ($D_2$), 15–30 cm ($D_3$), and 30–50 cm ($D_4$).

**Table 1.** Details of the sampling plots receiving irrigation water at different levels of salinity (mean $\pm$ SD, n = 10). Data were mean $\pm$ standard deviation (n = 10).

| Plot No. | Salinity Value (g/L$^{-1}$) | Location (Miles along the Desert Highway from Minfeng County) |
|---|---|---|
| $S_1$ | $23.80 \pm 1.12$ | 140 km + 470 m |
| $S_2$ | $18.10 \pm 0.84$ | 168 km + 800 m |
| $S_3$ | $13.99 \pm 0.66$ | 176 km + 000 m |
| $S_4$ | $8.90 \pm 0.45$ | 184 km + 500 m |
| $S_5$ | $5.75 \pm 0.29$ | 277 km + 500 m |
| $S_6$ | $2.58 \pm 0.13$ | 318 km + 100 m |

*2.2. Sampling*

The chosen points were sampled five times. Plant residues in the soil samples were discarded. The samples from the same depth but from different sampling points were pooled, evenly mixed, sieved through a 2 mm sieve, immediately stored in sterile sample bags in a car refrigerator at $-4\,°$C, and transported to the laboratory [13]. The samples for the determination of the physical and chemical properties, biomass, and enzyme activity were air-dried in the laboratory, and those for the determination of the bacterial diversity were stored in a refrigerator at $-20\,°$C and analysed within 4 weeks.

*2.3. Determination Methods*

2.3.1. Diversity of the Soil Bacterial Community

As a carrier to study the metabolic function of the environmental bacterial community based on a redox reaction, the Biolog Eco board has been widely used in the study of functional diversity of the soil bacterial community. The Biolog Eco method consisted of the preparation of buffers, extraction, inoculation, incubation, and determination [14].

The polymerase chain reaction–denaturing gradient gel electrophoresis (PCR–DGGE) consisted of DNA extraction, PCR amplification, denaturing gradient gel electrophoresis, gel extraction, DGGE profile analysis, DNA cloning, and sequence analysis [15]. We used the PfuTurbo Cx Hotstart DNA Cloning Kit (Agilent, Santa Clara, CA, USA) for the PCR. We applied a gradient from a 40-65% denaturing agent. The primer pair 16F27 (forward primer: 5′-AGA GTT TGA TCC TGG CTC AG-3′)/16R1522 (reverse primer: 5′-AAG GAG GTG ATC CAG CCG CA-3′) was applied to target the 16S rDNA V3 of the bacteria [16]. The cloning sequence analysis was performed by Shanghai Sangon Biotechnology Technology Service Co., Ltd, Shanghai, China.

The microbial phospholipid fatty acid analysis (PLFA) consisted of a fatty acid extraction, methylation, and identification [17].

2.3.2. Physical and Chemical Analysis

All the standard physical and chemical properties were determined [18]. The organic matter (OM) was determined by using the potassium dichromate volumetric method with external heating; total nitrogen (TN) by perchloric acid-sulphuric uric acid digestion; available nitrogen (AN) by alkaline hydrolysis distillation; total phosphorus (TP) by the acid dissolution-molybdenum antimony resistance colourimetric method; available phosphorus (AP) by the extraction-molybdenum antimony resistance colourimetric method; and total potassium (TK) by acid dissolution-flame photometry.

The soil moisture content was estimated by drying and the soil bulk was estimated by the cutting ring method. The total porosity was calculated from the soil volume, weight, and bulk [10].

### 2.3.3. Soil Bacterial Mass

The soil bacterial biomass was extracted by fumigation and leaching. Among them, the bacterial biomass carbon was fumigated and determined by the extraction–volumetric analysis method, the bacterial biomass nitrogen was fumigated and determined by the extraction–ninhydrin colourimetric method, and the bacterial biomass phosphorus was fumigated and determined by the extraction–total phosphorus determination method [19]. The kEc value used in the fumigation extraction method was 0.38.

### 2.3.4. Soil Enzyme Activity

Activity of the soil protease, cellulose, invertase, phosphatase, and urease was determined by colourimetry and that of the catalase activity by titration [20].

### *2.4. Data Analysis*

BIOLOG metabolism: average well colour development (AWCD)

$$AWCD = \Sigma\,(C - R)/n \tag{1}$$

(n differs with the type of medium [14]; in the present study, n was 31.)

PLFA profiles: gas chromatography–mass spectrometry (GC–MS) was used for the determination of the fatty acid species. A variance analysis of the fatty acid profiles was conducted using STATISTICA ver. 6.0 (StatSoft Inc., Tulsa, OK, USA).

DGGE fingerprinting: the gel interpretation and the calculation of mobility, intensity, and area were carried out using Bio-Rad's gel imaging system (Quantity One 4.4.1, Bio-Rad, Hercules, CA, USA).

Diversity index: the Shannon–Wiener diversity index equation, namely

$$H = -\sum_{i=1}^{S} P_i \ln p_i,\; E_{\mathrm{H}} = H/H_{\max} = H/\ln S \tag{2}$$

was used, where the diversity index ($H$), richness ($S$), and evenness ($E_{\mathrm{H}}$) represent the diversities of the soil bacterial carbon metabolism, fatty acids, and DNA segments, respectively.

Differences in soil microorganisms: a variance analysis and multiple comparisons with the least significant difference (LSD) method were performed using DPS ver. 9.5 (Data Processing System, Science Press, Beijing, China) to reveal whether the soil bacterial diversity differed significantly among different sites.

### *2.5. Evaluation of Soil Fertility*

The soil fertility reflects the combined physical, chemical, and biological properties of soil. To reliably reveal the differences in a single fertility index among different soil layers, the average values of the soil parameters at 0–5 cm, 5–15 cm, 15–30 cm, and 30–50 cm were analysed using the principal component analysis (PCA) to establish an evaluation system for soil fertility in the shelterbelt bordering the Tarim Desert Highway. In this evaluation system, the physical properties include the bulk density, total moisture content, and median diameter of the soil particles; the chemical properties include the soil nutrients (OM, TN, TP, TK, AN, AP, AK) and salinity-related parameters, including the pH, total salt content, and cation exchange capacity (CEC); and the biological properties include the bacterial diversity index (carbon source metabolism, species of fatty acid, and DNA segments), the activity of the soil enzymes (protease, cellulose, invertase, phosphatase, urease, and catalase), and the C, N, and P content of the soil bacterial biomass. The weights of all the fertility indicators were shown in Table 2.

**Table 2.** The weights of all the fertility indicators.

| Fertility Factor | Community | Fertility Factor | Community |
|---|---|---|---|
| Organic matter | 0.036 | DNA segments | 0.033 |
| Total nitrogen | 0.043 | Carbon source metabolism | 0.039 |
| Total phosphorus | 0.041 | Species of fatty acid | 0.038 |
| Total potassium | 0.037 | Catalase | 0.039 |
| Available nitrogen | 0.044 | Phosphatase | 0.040 |
| Available phosphorus | 0.041 | Urease | 0.043 |
| Available potassium | 0.041 | Cellulose | 0.033 |
| pH | 0.049 | Urease | 0.039 |
| Electrical conductivity | 0.034 | Protease | 0.036 |
| Total salt | 0.043 | Bacterial biomass carbon | 0.044 |
| Bulk density | 0.041 | Bacterial biomass nitrogen | 0.039 |
| Median diameter of soil particles | 0.040 | Bacterial biomass phosphorus | 0.041 |
| Moisture content | 0.047 | | |

The soil integrated fertility index (IFI), viewed as a synthetic index of each soil parameter, has been used in another study [21]. We selected the continuous function to evaluate each factor because each parameter varies continuously. The ascending or descending property of this function was determined by the positive or negative property of the factor loading of the principal component, which agrees with the effects of these factors on vegetation. For the bulk density, pH, and total salt content, a function with a descending property was chosen:

$$F(X_{ij}) = (X_{imax} - X_{ij})/(X_{imax} - X_{imin}) \qquad (3)$$

However, for the soil moisture content, porosity, and all the chemical and biological factors, a function with an ascending property was introduced:

$$F(X_{ij}) = (X_{ij} - X_{imin})/(X_{imax} - X_{imin}) \qquad (4)$$

In the above equations, $F(X_{ij})$ shows the relative importance of each fertility factor, $X_{ij}$ is the value of each fertility factor, and $X_{imax}$ and $X_{imin}$ express the maximum and minimum value of one factor.

Because the soil fertility parameters differ in their importance, a weight coefficient was introduced, which was obtained by calculating the percentage of the communality of one factor, and the numerical conversion was performed as follows:

$$W_i = Community_i / \sum_{n=1}^{i} Community_i \qquad (5)$$

According to the additive–multiplicative rule, the fertility index is synthesized multiplicatively by the individual parameters. The integrated fertility index is, therefore, calculated by the following equation:

$$IFI = \sum [W_i \times F(X_{ij})] \qquad (6)$$

where $W_i$ is the weight coefficient of each fertility factor and $F(X_{ij})$ expresses its relative importance.

## 3. Results and Analysis

### 3.1. Soil Bacterial Diversity

Table 3 shows that the carbon metabolic diversity index, genetic diversity index, and fatty acid diversity index of soil microorganisms decreased with an increase in salinity. The variance analysis and multiple comparisons indicated that the differences in the genetic diversity index $H$ and richness $S$ among the shelterbelt soils receiving irrigation water of different salinity levels were highly significant ($p > p_{0.01}$), and the differences among

the soils in terms of the diversity index $H$ and the richness index $S$ representing carbon metabolism were also significant ($p > p_{0.05}$), whereas those in terms of the diversity index $H$, richness index $S$, evenness index $E_H$ of fatty acids, and evenness index $E_H$ of genes were not significant ($p < p_{0.05}$). Thus, the genetic diversity of soil microorganisms was most affected by the differences in the salinity, followed by the carbon metabolic diversity and fatty acid diversity, in that order, suggesting that the species distribution evenness indexes of soil bacterial genes and of fatty acids were not particularly affected. Therefore, when the salinity was high, the metabolic activity of soil microorganisms was low and so were the diversities of the gene and fatty acid species.

**Table 3.** Changes in the soil bacterial diversity indexes due to the salinity levels (n = 5, Mean ± SE).

| Plot | Carbon Mechanism | | | Genes | | | Fatty Acids | | |
|---|---|---|---|---|---|---|---|---|---|
| | $H$ | $S$ | $E_H$ | $H$ | $S$ | $E_H$ | $H$ | $S$ | $E_H$ |
| $S_1$(23.80) | 2.85 ± 0.23 a | 21.75 ± 2.84 a | 0.93 ± 0.12 a | 0.82 ± 0.07 A | 12.34 ± 1.61 A | 0.25 ± 0.02 a | 1.30 ± 0.35 a | 20.08 ± 1.83 a | 0.33 ± 0.05 a |
| $S_2$(18.10) | 3.14 ± 0.47 b | 25.75 ± 3.16 b | 0.97 ± 0.14 a | 0.87 ± 0.06 A | 12.96 ± 1.80 A | 0.29 ± 0.04 a | 1.47 ± 0.38 a | 21.69 ± 2.48 a | 0.41 ± 0.08 ab |
| $S_3$(13.99) | 3.20 ± 0.35 b | 27.13 ± 3.60 c | 0.97 ± 0.17 a | 1.00 ± 0.06 A | 14.25 ± 2.57 A | 0.39 ± 0.04 b | 1.51 ± 0.33 ab | 23.5 ± 5.07 ab | 0.39 ± 0.09 a |
| $S_4$(8.90) | 3.24 ± 0.39 b | 27.88 ± 2.82 c | 0.97 ± 0.18 a | 1.34 ± 0.32 B | 20.25 ± 3.18 B | 0.50 ± 0.11 c | 1.65 ± 0.60 b | 25.75 ± 2.56 b | 0.55 ± 0.15 bc |
| $S_5$(5.75) | 3.26 ± 0.44 b | 28.63 ± 4.58 c | 0.97 ± 0.19 a | 1.78 ± 0.54 C | 35.00 ± 5.34 C | 0.48 ± 0.10 c | 1.73 ± 0.75 bc | 30.01 ± 4.91 c | 0.58 ± 0.11 c |
| $S_6$(2.58) | 3.59 ± 0.45 c | 29.13 ± 4.22 c | 1.06 ± 0.21 a | 2.43 ± 0.47 D | 41.75 ± 5.9 D | 0.63 ± 0.13 d | 1.86 ± 0.92 c | 32 ± 8.42 c | 0.57 ± 0.14 c |
| *p* value | * | * | | ** | ** | | | | |

Note: (1) The least significant difference (LSD) method was used for determining the significance (F test); n = 5; ** and * denote significance levels of 0.01 and 0.05, respectively; and capital and small letters after numbers denote highly significant and significant differences, respectively. (2) $H$—Diversity index, $S$—Richness index, $E_H$—Uniformity index.

According to Table 4, the homology value of the bacteria was greater than 95% after a phylogenetic comparison on the GenBank database. Among the 30 soil samples were *Janthinobacterium* sp., *Thiobacillus denitrificans*, *Acidovorax* sp., *Shewanella frigidimarina*, and *Anaeromyxobacter*. In more than half of the soil samples, the homogeneity of the electrophoretic bands with *Janthinobacterium* sp. and *Thiobacillus denitrificans* reached 100%.

**Table 4.** Tentative identification of the dominant DGGE bands by sequencing the excised and BLAST analysis.

| Bands | Closest Species in the GenBank Database | Sequence Accession Number | Similarity (%) |
|---|---|---|---|
| C1 | *Janthinobacterium* sp. | AY88012 | 100 |
| C2 | *Acidovorax* sp. | AY88013 | 99 |
| C3 | *Anaeromyxobacter* | AY88014 | 96 |
| C4 | *Thiobacillus denitrificans* | AY88015 | 100 |
| C5 | *Shewanella frigidimarina* | AY88016 | 97 |

Note: C1, C2, C3, C4, C5 refers to the five bands marked in the DGGE electrophoretic picture after being cut.

### 3.2. Soil Fertility

As can be seen in Figure 1, as the salinity decreased, the IFI increased significantly ($F = 71.21 > F_{0.01}$): at the highest level of salinity ($S_1$, 23.8 g/L), the IFI was less than 0.25; at the next level ($S_2$, 18.10 g/L), the IFI was slightly greater than 0.3; the trend continued until at the lowest salinity ($S_6$, 2.58 g/L), the IFI exceeded 0.65, a value that was more than double that at $S_2$. Therefore, highly saline water for irrigation is not conducive to soil fertility.

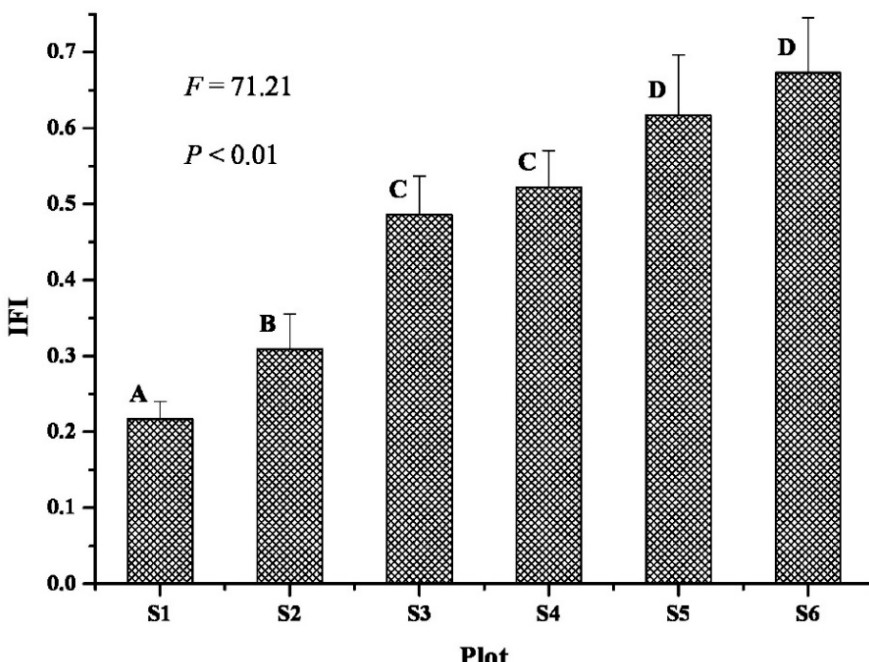

**Figure 1.** Soil fertility following irrigation with water of different salinity levels (S1, S2, S3, S4, S5, S6 refer to the plots with different salinity value mentioned in Table 1; A, B, C, D represents the result of a letter marker in a variety of comparisons).

The soil fertility also varied with depth, with the deeper layers being more fertile. Figure 2 shows that the IFI of the top layer ($D_1$, 0–5 cm) was the lowest (less than 0.3) and was significantly lower than that of any other layer. However, among the three other layers, the difference in fertility was not significant.

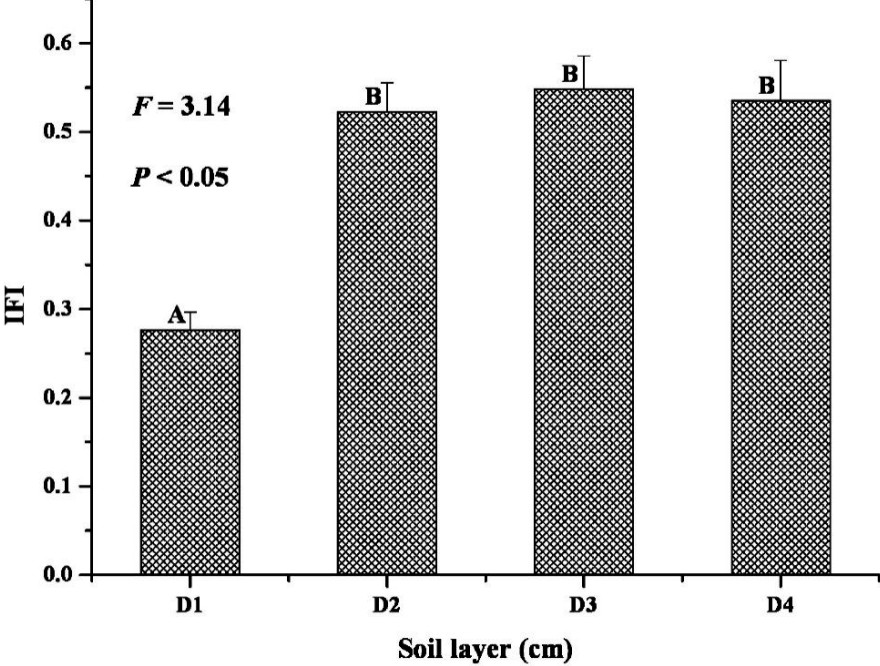

**Figure 2.** Soil fertility at different depths (D1, D2, D3, D4 refer to the different soil layer; A, B represents the result of a letter marker in a variety of comparisons).

### 3.3. Relationship between Soil Bacterial Diversity and Soil Fertility

Figure 3 shows the power function relation between the AWCD values, which represent the carbon metabolic activity of soil microorganisms, and the IFI. The function equation was $Y = 0.56X^{0.66}$ and the coefficient of determination $R^2$ was 0.82. Therefore, the IFI would increase proportionally to the power of the soil carbon metabolic activity.

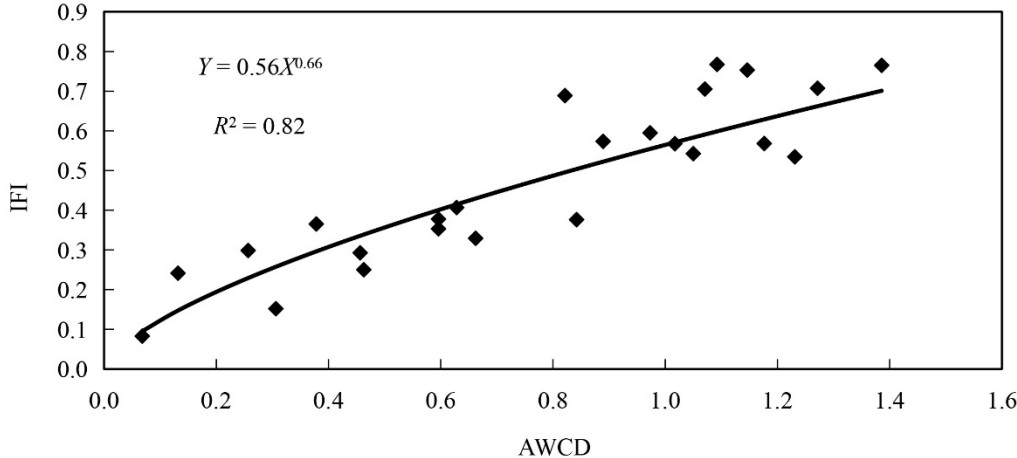

**Figure 3.** The relationship between the metabolic diversity of soil microorganisms and soil fertility.

Figure 4 shows the power function relation between the diversity index $H$ of fatty acids in soil microorganisms and the IFI. The function equation was $Y = 2 \times 10^{-7}X^{15.35}$ and the coefficient of determination $R^2$ was 0.889, indicating that the IFI increased proportionally to the power of the diversity index $H$ of the soil fatty acids.

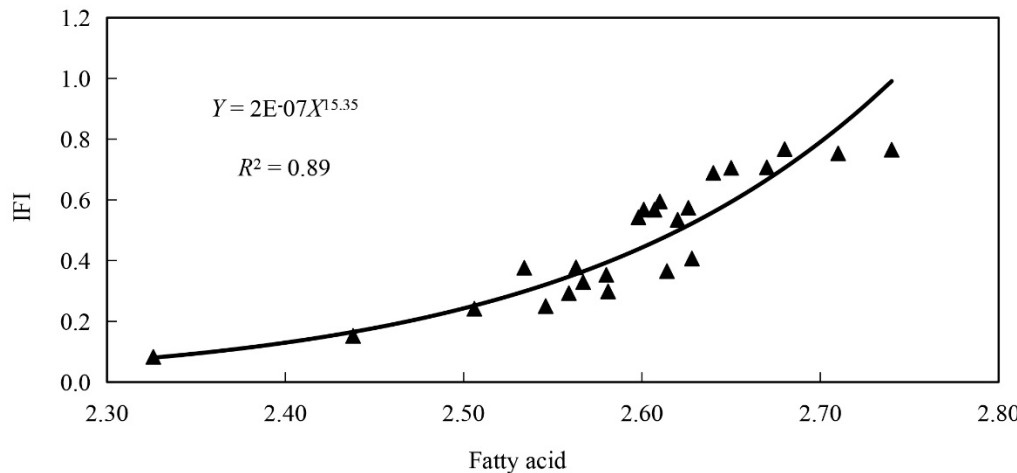

**Figure 4.** The relationship between the fatty acid diversity of soil microorganisms and soil fertility.

Figure 5 shows the polynomial relation between the diversity index $H$ of DNA segments of soil microorganisms and the IFI. The function equation was $Y = 1.07X^2 + 4.28X - 3.41$ and the coefficient of determination $R^2$ was 0.866, indicating that the IFI increased proportionally to the polynomial of the diversity index $H$ of the soil DNA segments.

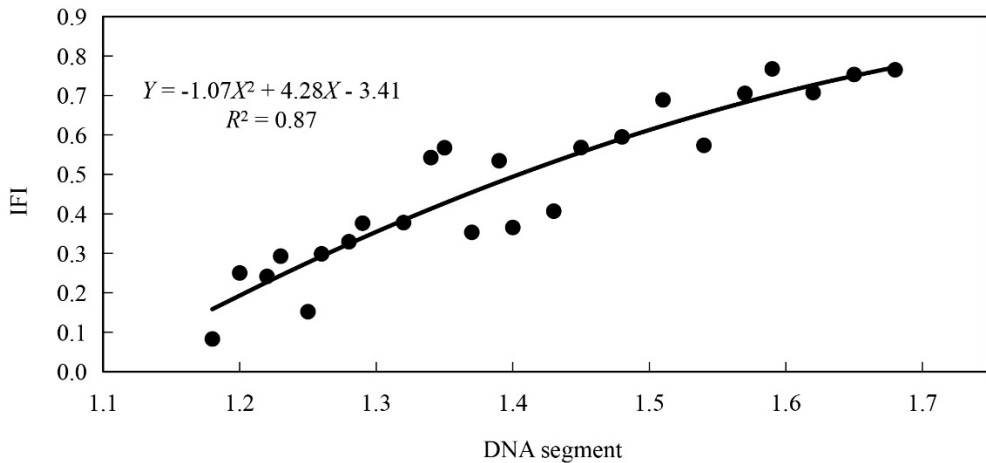

**Figure 5.** The relationship between the genetic diversity of soil microorganisms and soil fertility.

### 3.4. Composition Characters of Soil Salinity

Table 5 shows that the contents of salt ions in soils differed due to the level of salinity of the irrigation water. The ions, in descending order, were $SO_4^{2-}$, $Cl^-$, $Na^+$, $Ca^{2+}$, $HCO_3^-$, $K^+$, $Mg^{2+}$, and $CO_3^{2-}$. Three salt ions, $SO_4^{2-}$, $Cl^-$, and $Na^+$, together contributed more than 80% of the total, whereas $CO_3^{2-}$, at less than 1%, contributed the least. This distribution indicates that the major salts in the soils of the Tarim Desert Highway shelterbelt were $Na^+$, $Cl^-$, and $SO_4^{2-}$. The total soil salinity increased with an increase in the level of salinity of the irrigation water, although the $Na^+$ and $Cl^-$ contents decreased and the $SO_4^{2-}$ content increased.

**Table 5.** Contents (grams/litre) of eight main salt ions in soils irrigated with water at different salinity levels.

| Plot | $CO_3^{2-}$ | $HCO^{3-}$ | $Cl^-$ | $SO_4^{2-}$ | $Ca^{2+}$ | $Mg^{2+}$ | $Na^+$ | $K^+$ |
|------|------|------|------|------|------|------|------|------|
| $S_1$ | 0.09 | 0.31 | 6.36 | 6.04 | 2.10 | 0.58 | 3.47 | 0.32 |
| $S_2$ | 0.07 | 0.28 | 6.04 | 5.75 | 1.86 | 0.47 | 3.29 | 0.27 |
| $S_3$ | 0.01 | 0.12 | 4.46 | 4.88 | 1.52 | 0.26 | 2.97 | 0.12 |
| $S_4$ | 0.01 | 0.12 | 1.69 | 3.33 | 0.87 | 0.25 | 1.18 | 0.14 |
| $S_5$ | 0.03 | 0.17 | 1.31 | 0.71 | 0.14 | 0.07 | 0.92 | 0.11 |
| $S_6$ | 0.02 | 0.13 | 1.09 | 0.86 | 0.19 | 0.06 | 0.80 | 0.11 |

## 4. Discussion and Conclusions

### 4.1. Effect of Salinity of Irrigation Water on Soil Bacterial Diversity

The soil samples in this study were collected from non-rhizospheric areas of the shelterbelt trees; therefore, the effects of plants on soil salinity can be safely ignored. The soil salinity depended mainly on the salt content of the irrigation water and was thus related to the level of salinity of that water [22].

The soil bacterial diversity of the Tarim Desert Highway shelterbelt differed significantly with the level of salinity of the irrigation water. A higher salinity exposes the soil microorganisms to greater salt stress, which lowers not only the diversity of the bacterial populations but also their activity [23]. The soil salinity affects the availability of soil nutrients; for example, a massive accumulation of $Ca^{2+}$ results in more of the soil phosphorus being fixed, thereby reducing its availability to plants [24]. The richness and composition of the soil nutrients determine the species, numbers, and activities of the microorganisms [25]. In the Tarim Desert Highway shelterbelt, the irrigation salinity mainly affected the soil salinity in the 0–20 cm range [26]. Therefore, highly saline water used for drip irrigation is not conducive to the survival of microorganisms and adversely affects their functional diversity, genetic diversity, and species diversity in the Tarim Desert Highway shelterbelt.

*4.2. Changes in Soil Fertility under Drip Irrigation with Saline Water*

The IFI values increased significantly with a decrease in the salinity of the drip irrigation. Highly saline water used for drip irrigation is not conducive to soil fertility. The salt accumulation caused greater bulk density, which is not conducive to the normal turnover of soil nutrients and their availability, resulting in an overall decline in the soil quality [27].

The differences in soil fertility at different depths were obvious. The lower fertility of the top layer was due to severe soil compaction, and the topsoil was also affected by sandstorms. In fact, most of the topsoil consisted of mobile aeolian sand, which directly lowers its fertility, whereas the lower layers were protected in one sense and hence more fertile [28].

Gu et al. [29] suggest that the use of saline water for irrigating the extremely arid areas that border the Tarim Desert Highway helped not only in establishing the shelterbelt, but also in making the aeolian sandy soils more fertile by improving their physical and chemical properties. Wang et al. [10] found that the nutrient content of soils from a four-year-old plantation of Ephedra was much higher than that of a site along artemisia ordosicaartemisia ordosicaartemisia ordosicaartemisia ordosicathe southern edge of the Mu Us Sandland. Cao et al. [11] found that the physical and chemical properties of soils were improved after a plantation of *Caragana microphylla* Lam. was established in the sand dunes of the Horqin Sandland, and that the activity of soil enzymes increased gradually with the age of the forest. Therefore, establishing or restoring vegetation, even by using saline water for drip irrigation, is good for the soil—it increases the soil nutrient levels and encourages biological activity.

*4.3. The Relationship between Soil Microorganisms and Soil Fertility*

Microorganisms play an important role in maintaining soil fertility. They convert naturally occurring nutrients in soil and those added from fertilizers from unavailable forms to available forms. The soil ecosystem is primarily controlled by soil microflora, and small changes in soil can lead to large changes in the bacterial diversity [30]. Changes in the soil bacterial communities, which are governed by soil fertility and environmental conditions, can be used as important biological indicators of soil fertility [31].

Therefore, using less saline water for drip irrigation could avoid salt accumulation in soil and arrest its compaction, promote the formation of soil aggregates and the build-up of nutrients, and increase microbial activity, thus playing a crucial role in promoting the circulation, conversion, and utilization of nutrients in aeolian sandy soils and improving the soil quality. The judicious use of saline water, therefore, deserves serious consideration in irrigation practices.

**Author Contributions:** Data curation, H.W.; writing—original draft preparation, X.Y.; writing—review and editing, Z.J. All authors have read and agreed to the published version of the manuscript.

**Funding:** This research was funded by the Natural Science Foundation of China, grant number 41571498.

**Institutional Review Board Statement:** Not applicable.

**Informed Consent Statement:** Not applicable.

**Data Availability Statement:** The data that support the findings of this study are available from the corresponding author upon reasonable request.

**Acknowledgments:** The authors wish to thank Key Laboratory of Biogeography and Bioresource in Arid Land, Xinjiang Institute of Ecology and Geography, Chinese Academy of Sciences, and Xinjiang Corps' Oasis Agriculture Key Laboratory for providing technical supports.

**Conflicts of Interest:** The authors declare no conflict of interest.

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
