# Peer review of "Effect of Saline Water for Drip Irrigation on Microbial Diversity and on Fertility of Aeolian Sandy Soils"

_diversity, doi:10.3390/d13080379_

Round 1

Reviewer 1 Report

This manuscript analyses  the impact of salinity on soil microbial diversity and soil fertility in a desert setting. Molecular techniques are used to assess the microbial diversity.

Comments:

Please, give the common name along with the Latin binomial of all the plants mentioned in the text. This is common practice in scientific papers.

Line 57: “…conventional methods” means different things to different people. Please, expand the explanations to make clear what the authors understand for “conventional”

Line 76 – what trees? Not clear what relevance the trees have here. Also, explain the study site in more detail: what is it used for? Why is it relevant to study this site? What is the aim of the ‘belt’? Is it to grow crops, or to grow salt-resistant shrubs? Please clarify so that the relevance of the study can be understood.

Overall, the Methods are not clearly written. They should be explained so that they can be replicated by any future researchers.

Lines 91-97 – explain what microbial diversity is being targeted in the study, e.g. bacteria, archaea, protozoa, protists in general.

Line 109: explain “fumigation”. Not clear as it is, or why fumigation is relevant in this context.

Table 2 needs to include the salinities so that the data are clearer

Overall, the paper demonstrates that there seems to be an impact of increased salinity on soil microbial diversity and fertility at this site (but see below).  Unfortunately, there is no differentiation or explanations about what types of microbes are being targeted, which would help direct future research on that topic. Direct microscope observations in this case would have been very valuable in order to distinguish active from non-active microbes, particularly the eukaryotic microbes.  Targeting genes only may give a wrong measure of microbial diversity since it is possible that many microbes might actually be inactive (i.e. dormant or in cysts) but their genes are still being isolated with the methodology applied.

There are some relevant papers, e.g. Is the Taklimakan Desert Highway Shelterbelt Sustainable to Long-Term Drip Irrigation with High Saline Groundwater? Jianguo Zhang, Xinwen Xu, Shengyu Li ,Ying Zhao, Afeng Zhang, Tibin Zhang, Rui Jiang, 2016, https://doi.org/10.1371/journal.pone.0164106, that are not cited but are very relevant to the research presented here as they also deal with salinity studies at the sampling location, and whose research contrasts to the one presented in this manuscript, especially the results after the 20cm soil depth. These contrasting results should be discussed.

Author Response

Please see attachment below. 

Reviewer 2 Report

The manuscript D diversity-1170254 entitled Effect of saline water for drip irrigation on microbial diversity and on fertility of aeolian sandy soils is in line and scope of Diversity and could be publish in the journal, but minor revision is needed.

Abstract:

There is no posture hypothesis at the beginning, which is to be tested later in the article.

Introduction:

‘Since soil microflora is the most potential and sensitive indicator of soil fertility, it is important to study the effects of changes in soil fertility on microbial diversity.’ – it is perpetuum mobile, please rewrite.

Results:

Table 1 salinity – means and standard deviations are lacked.

Table 2 – please add an explanation under the table what means uppercase and lowercase letters.

Discussion:

Lines 251-256 – sentence too long, please split it.

Lines 257-262 – unclear.

‘Changes in soil microbial communities, which are governed by soil fertility and environmental conditions, can be used as important biological indicators of soil fertility’ – What is the point of carrying out expensive microorganism analyzes since fertility can be measured directly?

I am afraid that in desert areas the use of even light brine is risky and brings positive effects only for a short time, and then leads to soil salinity. This stipulation should be included in the final section and discussed with similar and long-term research carried out elsewhere in the world.

Author Response

Please see attachment below. 

Reviewer 3 Report

diversity-1170254-peer-review-v1

The present study investigates the effect of soil irrigation with saline water on the microbial community in terms of sequence heterogeneity, metabolic activity and fatty acid composition. The applied methods are quite simple but are suitable to answer the scientific questions.

However, although the research question is interesting, the current presentation of the results is not convincing and hard to interpret because basic methodological information is missing. For example, the authors did not state what kind of sequence they investigated in their qPCR-DGGE approach. Moreover, it is not clear, if the calculated fertility index is a novelty of the present study or has been established before.

The introduction is missing information about research of irrigation with saline water from other habitats, as well as statements about the importance of this study.

The representation of the results focuses too much on the calculated IFI, which, in return, has not been well explained in the methods section. Little attention has been given on the investigation of real relationships between salinity and microbial community structures.

Taken together, I cannot recommend publication of the article in its current form.

Below you can find some more detailed information:

Abstract:

l17: remove full stop

INTRODUCTION:

The introduction misses a paragraph explaining the irrigation with saline water: is this a common practice, why is saline water used? how salty is it? Advantages and disadvantages. etc.

l25: I suggest to remove „forest“ as it is not only important for forests

l31: the term “microflora” is generally obsolete and misleading as it is not speaking of any plants or “flora”. I suggest to change to microbiome or something similar.

l34: how is the shelter belt contributing to a safe operation of the highway? Please include a sentence describing its function (i.e. preventing sand transport on the street or similar)

l37: surround instead of surrounds

l58-62: Change to: The following hypotheses were addressed: (i) saline water is significantly altering soil microbial diversity soil fertility in the shelter belt and (ii) soil microbial diversity and soil fertility are depending on irrigation water salt concentrations and on soil depth.

M&M

l71-72: it is not clear to me, if this belt is regularly irrigated with salty water, of if this irrigation was only done for the purpose of this experiment. If it is irrigated also under normal conditions, it would be good to mention what kind of water is being used to irrigate the belt normally.

l73-75: these mentioned variables are not included in table 1, but it would be good if they were. Also, I am not sure what water well no. in table 1 means and if it is important to include this information. Rather, information such as the once mentioned in l 74-76 would be important to include.

2.3. Determination methods: This chapter is too general. More information needs to be included for the single methods. E.g. what Primer where used for the PCR? What gradient did you choose for the DGGE? What cloning kit was used? How was sequence analysis performed? What kind of machines were you using? References for physical and chemical analyses? What Kec-value was used in the fumigation extraction method etc…

It is clear that each method has its reference, but to have an idea of the achieved results, the reader needs to get this crucial information.

l91-92: For those unfamiliar with the biology-eco method, please include a sentence about the principle of this method.

l134: The construction of IFI is not clear to me. Has it been done in other publications before, or are you establishing it? As I understood, it contains all of your measured variables with a certain individual weighting. The weighing, however, is unclear. In order to give a better overview on this IFI, I suggest to show the results of the PCA at least in the supplementary data, and to generally include more information in the text.

l144: Reference 23 is not about an integrated fertility factor, or at least it is not apparent from the abstract. Please give more information on how it is formed and where it has been studied already.

l156: The percentage of communality of a one factor is not clear to me. What are you summing up when you sum up “Community”? Please explain more clearly.

l127-129: As you are comparing more than two sample groups at a time, I suggest to compute an ANOVA and to include p-values instead of F-values, as this is more commonly done and will be familiar to a greater readership.

RESULTS
Table2: Why is Carbon mechanism showing AWCD, but not EH values?

l181: please change to ”highly significant”

FIG3: Wasn’t the IFI constructed ALSO including the AWCD values? If this is the case, it is logical that these two factors are correlated somehow, and you are just showing their autocorrelation. Same would apply for Fig. 4 and Fig. 5. These three figures are not directly contributing to your research question, as they do not investigate the effect of depth or salinity on those factors. Instead, I suggest to investigate more these particular relationships and to show e.g. the relationship of salt concentrations on the single microbial variables (genetic, fatty acids, biology). This might give more information on how salt concentrations might influence microbial communities, while the information shown in fig. 3-5 is quite general (higher diversity means higher fertility).

FIG 6: is rather descriptive and might better be shown in absolute values and in a table. Like this, it seems that all salinity classes (S1-S6) contained the same amounts of salt ions, although it is just the relative abundance that stayed comparable.

DISCUSSION:

Given that an updated version of the article may differ significantly from the present manuscript, I do not go into details for this chapter.

Author Response

Please see attachment below.

Round 2

Reviewer 1 Report

The authors have attended some of my concerns but not fully.

Please, give the common name along with the Latin binomial of all the plants mentioned in the text, not just the Latin names. This is common practice in scientific papers.

The Methods and the overall text is still unclear.  If bacteria are the only microorganisms that have been targeted in this manuscript (MS), it should clearly indicate so. The MS keeps mentioning “soil microbial community”, but the only microorganisms that have been investigated (according to the authors’ response to my first round of comments) are bacteria. This should be clearly stated throughout the MS, e.g. “soil bacterial community”.  Likewise when describing the “soil microbial biomass” in the Methods, or soil microbial diversity in the Discussion.

Author Response

The authors have attended some of my concerns but not fully.

Reply: we are very grateful to you good suggestions.  We are sorry for not providing you all the replies.

Please, give the common name along with the Latin binomial of all the plants mentioned in the text, not just the Latin names. This is common practice in scientific papers.

Reply: we are sorry for not providing the Latin binomial of all the plants, we have provided them as below: Haloxylon ammodendron (C. A. Mey.) Bunge, Calligonum mongolicum Turcz., Tamarix chinensis Lour., Caragana microphylla Lam., Artemisia ordosica Krasch.

The Methods and the overall text is still unclear.  If bacteria are the only microorganisms that have been targeted in this manuscript (MS), it should clearly indicate so. The MS keeps mentioning “soil microbial community”, but the only microorganisms that have been investigated (according to the authors’ response to my first round of comments) are bacteria. This should be clearly stated throughout the MS, e.g. “soil bacterial community”.  Likewise when describing the “soil microbial biomass” in the Methods, or soil microbial diversity in the Discussion.

Reply: we are sorry for not stating the experimental method clearly , and we have modified ‘microbial’ instead of ‘bacterial’ throughout this MS.

Reviewer 3 Report

see pdf

Author Response

Unfortunately, the manuscript has not been revised in an appropriate way, as many of my suggestions have not been implemented, or the authors couldn’t convince me why the changes have not been done.

Some major points:

Still, more information about the applied techniques needs to be included (see below)

Moreover, it is STILL not clear, if the calculated fertility index is a novelty of the present study or has been established before.

STILL, little attention has been given on the investigation of real relationships between salinity and microbial community structures, as outlined in my first comment on figures 3-5. This should be seriously addressed prior to the publication of the manuscript.

Specific points (Discussion not included):

ABSTRACT

The first two new sentences contain a number of errors: A suggestion: Saline water is widely distributed in ... and sometimes represents the only source of irrigation water to restore and reconstruct vegetation. However, the effects of saline water on … In this study we investigated a vegetation belt along the Tarim Desert Highway that has been

constructed as wind break and consists of desert shrubs and was irrigated with saline water at six levels of salinity.

Reply: we have revised the two sentences following your suggestions.

INTRODUCTION

L51: remove comma

Reply: we have removed the comma.

L52: write in italics the names of the shrubs. (here and throughout the manuscript)

Reply: we have checked the names of the shrubs and wrote them in italics throughout the manuscript.

L52-55: I suggest to shorten this sentence by adding it to the previous one: … salt and alkali and can only be kept alive by drip irrigation with the available salty groundwater.

Reply: we are very sorry since we didn’t understand your point.

M&M

L94-96: This sentence is a repetition of the previous sentence and should be deleted (including the sentence about drip irrigation).

Reply: we are sorry for having not found the previous sentence.

L103: I suggest to add : …and tree species (only Haloxylon ammodron), but different ground water salinity levels.

Reply: we are sorry for not understanding your point.

TABLE1: I am sorry for the confusion, I thought that all these variables were specifically recorded for the sites and therefore suggested to include the actual values in the table. As in your case you kept those variables constant it is better to re-integrate the information in the text as in your original version, and remove the note of the table.

Reply: we are very grateful for your suggestion and deleted the values and the note of the table 1.

Determination methods: This chapter is STILL too general, as almost no further information was included. I do not request detailed information about all methods, but at least the following statements (its no more than some 10 words more per statement).

Reply: we are very grateful for your suggestion and we'll try to follow your suggestion.

E.g. what Primer where used for the PCR (e.g.: using the primer pair XX/XY targeting the XX gene of microbe XX(Ref))?

Reply: we have briefly introduced the PCR by the sentence “using the primer pair 16F27 (forward primer: 5'-AGA GTT TGA TCC TGG CTC AG-3') / 16R1522 (reverse primer: 5'-AAG GAG GTG ATC CAG CCG CA-3') targeting the 16S rDNA V3 of bacteria (Pennanen T, Paavolainen L, Hantula J. Rapid PCR-based method for the direct analysis of fungal communities in complex environmental samples [J]. Soil Biology and Biochemistry, 2001, 33: 697-699).”

What gradient did you choose for the DGGE (… applying a gradient from xx-xy% denaturing agent)?

Reply: we have provided the gradient of denaturing agent, by the sentence “applying a gradient from 40-65% denaturing agent”.

What cloning kit was used (mention cloning kit and company or the reference for your protocol)?

Reply: we used PfuTurbo Cx Hotstart DNA Cloning Kit (Agilent, USA) for PCR.

How was sequence analysis performed (i.e. sanger sequencing of PCR products? by a certain company, or with a certain machine in your lab? )

Reply: The cloning sequence analysis was performed by Shanghai Sangon Biotechnology Technology Service Co., Ltd, China.

References for physical and chemical analyses?

Reply: we have provided the reference [Institute of Soil Science, Chinese Academy of Science. Soil physical and chemical properties analysis, Science and Technology Press in Shanghai, Shanghai, China,  1978. ] for soil physical and chemical analyses.

What Kec-value was used in the fumigation extraction method etc…

Reply: the Kec-value used in the fumigation extraction method was 0.38.

Reference to Table 2 has to be given at some point in the text

Reply: we have referenced Table 2 in the text.

IFI: I still don’t know if this IFI has been calculated by someone else before, or if it has been done for the first time in this study. Please include this specific information. If you did it, no problem, but you need to state this clearly.

Reply: Soil integrated fertility index (IFI), as be viewed a synthetic index of each soil parameter, has been used in other study [23].

REFERENCES: Numbering is not correct anymore.

Reply: we have corrected the numbering of reference throughout the text.

DISCUSSION

Given that an updated version of the article may differ significantly from the present manuscript, I do not go into details for this chapter.

Reply: We have improved the quality of this section.
